# Redefining innate natural antibodies as important contributors to anti-tumor immunity

Kavita Rawat[1], Anita Tewari[1], Madeline J Morrisson[1], Tor D Wager[2], Claudia V Jakubzick[1]*

[1]1Department of Microbiology and Immunology, Dartmouth Geisel School of Medicine, Hanover, NH, United States; [2]Department of Psychological and Brain Sciences, Dartmouth Geisel School of Medicine, Hanover, NH, United States

**Abstract** Myeloid, T, and NK cells are key players in the elimination phase of cancer immunoediting, also referred to as cancer immunosurveillance. However, the role of B cells and NAbs, which are present prior to the encounter with cognate antigens, has been overlooked. One reason is due to the popular use of a single B cell-deficient mouse model, muMT mice. Cancer models using muMT mice display a similar tumor burden as their wild-type (WT) counterparts. Empirically, we observe what others have previously reported with muMT mice. However, using two other B cell-deficient mouse models (IgHELMD4 and CD19creDTA), we show a three- to fivefold increase in tumor burden relative to WT mice. In addition, using an unconventional, non-cancer-related, immune neoantigen model where hypoxic conditions and cell clustering are absent, we provide evidence that B cells and their innate, natural antibodies (NAbs) are critical for the detection and elimination of neoantigen-expressing cells. Finally, we find that muMT mice display anti-tumor immunity because of an unexpected compensatory mechanism consisting of significantly enhanced type 1 interferon (IFN)-producing plasmacytoid dendritic cells (pDCs), which recruit a substantial number of NK cells to the tumor microenvironment compared to WT mice. Diminishing this compensatory pDC-IFN-NK cell mechanism revealed that muMT mice develop a three- to fivefold increase in tumor burden compared to WT mice. In summary, our findings suggest that NAbs are part of an early defense against not only microorganisms and dying cells, but precancerous cells as well.

*For correspondence: claudia.jakubzick@dartmouth.edu

**Competing interest:** The authors declare that no competing interests exist.

## Introduction

Tumorigenesis occurs when normal cells accumulate mutations and proliferate out of control. Extrinsic tumor suppression mechanisms (i.e., cancer immunoediting) arise after intrinsic tumor suppression mechanisms fail. Extrinsic tumor suppression mechanisms are led by immune cells and divided into three stages: elimination, equilibrium, and escape (*Schreiber et al., 2011*; *Mittal et al., 2014*). This study focuses on the elimination phase, particularly the innate recognition of neoantigen-expressing cells by natural antibodies (NAbs).

In the elimination phase of cancer immunoediting, the identification and destruction of nascent tumors result from the expression of neoantigens by mutant cells, which form a hypoxic cluster and release damage-associated molecular patterns (DAMPs). T cells and NK cells are the predominant immune cells to recognize these cellular abnormalities (*Rao et al., 2019*). To induce antigen-specific effector T cells against neoantigens (as with other foreign antigens), dendritic cells (DCs) must become licensed through their pattern recognition receptors by either DAMPs or pathogen-associated molecular patterns (PAMPs). This licensing allows for DCs to present neoantigens in an immunogenic fashion to T cells (*Ardavín et al., 2004*). Hence, in the elimination phase, DAMPs are the designated

mediators that license DCs to present neoantigens in an immunogenic fashion to T cells (*Spörri and Reis e Sousa, 2005*; *Desch et al., 2014*). However, newly transformed cells that just escaped intrinsic suppression mechanisms do not express PAMPs but must be eliminated too. So, how do immune cells become licensed to eliminate newly transformed neoantigen-expressing cells prior to cluster formation and hypoxic conditions?

This question, along with two observations in unconventional, non-cancer-related, immune neoantigen (i.e., new antigens to the host) models led us to hypothesize that NAbs significantly contribute to early neoantigen recognition. The first observation is the rejection of male cells in syngeneic female mice. This immune rejection is due to the presence of Y chromosome antigens, H-Y antigens, which are perceived as neoantigens to syngeneic female mice (*Atif et al., 2018*). The second observation is not H-Y antigen based. Adoptive transfer of MHC-matched 129Sv female cells into C57BL/6 female mice results in the rejection of 129Sv cells, and vice versa (*Atif et al., 2015*; *Carmi et al., 2015*; *Atif et al., 2018*). This immune response is due to the allelic variations outside of the MHC locus. Interestingly, what these two models share in common is that the adoptively transferred cells have normal MHC class I levels and no PAMPs present (*Sengupta et al., 2006*). Therefore, if there are no PAMPs, or clustering of cells leading to hypoxic conditions, how then are these neoantigen-expressing cells (i.e., C57BL/6 male and 129 female cells in C57BL/6 female mice) identified and subsequently cleared? Specifically, what is licensing endogenous antigen-presenting cells to present these neoantigen-expressing cells in an immunogenic fashion to T cells? In both models, we previously demonstrated that NAbs are required for this elimination process in a chain-link reaction: antibody bound neoantigen-expressing cells license tissue trafficking monocytes and CD11c+Irf4-dependent cells to present cell-associated neoantigens to CD4 T cells, which in turn license DC1 via CD40 to cross-present neoantigens and cross-prime CD8 T cells (*Atif et al., 2018*). In these non-cancer neoantigen models, NAbs are required for the recognition and rejection of neoantigen-expressing cells.

Currently, NAbs are thought to have no role in cancer immune surveillance. In large part, because a series of widely cited studies found that B cell-deficient muMT mice mount a robust anti-tumor immune response similar to or greater than wild-type (WT) (*Supplementary file 1*) mice. However, several converging lines of evidence challenge this view. At least three other, lesser-known models of B cell-deficient mice show markedly enhanced tumor growth (*Supplementary file 1*). Here, we confirm previous findings in one of the models, IgHEL mice, and examine tumor burden in a new B cell-deficient model, CD19creDTA mice. Furthermore, other methods for studying the role of B cells, such as antibody depletion with anti-CD20 or anti-IgM, show increased tumor growth, contradicting the muMT findings and suggest that B cells may mediate early anti-tumor immunity (*Supplementary file 1*). Overall, the discrepancies in the literature, along with the observations that NAbs are required to initiate the rejection of syngeneic male and MHC-matched 129Sv female cells in C57BL/6 (BL6) female mice, led us to hypothesize that NAbs may have similar functions in cancer immunosurveillance.

## Results

### NAb repertoire is essential for the elimination of neoantigen-expressing cells

First, we examined whether NAbs are required for the elimination of non-cancerous neoantigen-expressing cells. To investigate this, CD45 congenic cells were generated to track the acceptance or rejection of adoptively transferred cells into a CD45.2 host. Specifically, neoantigen-expressing CD45.1/2 129/BL6 cells, along with an internal control, CD45.1 BL6 cells, were adoptively transferred into CD45.2 BL6 mice (*Figure 1A*, experimental diagram). At 18 days post-transfer, WT and *Aicda*−/− mice (a hyper-IgM mouse) mounted a robust cytolytic response against 129/BL6 cells, resulting in its rejection (*Figure 1A*). In contrast, mice lacking an antibody repertoire, IgHEL mice (hypo-IgM), did not reject the neoantigen-expressing 129/BL6 cells (*Figure 1A*; *Muramatsu et al., 2000*). This finding suggests that NAbs are required for the elimination of neoantigen-expressing cells.

To assure that the HEL mice can mount a cytotoxic T cell (CTL) response, WT and IgHEL mice were immunized with OVA and poly I:C, a TLR3 ligand. This immunization is independent of antibodies, as the poly I:C directly licenses TLR3+ DC1 to present OVA as an immunogen to cognate T cells. Six days post-immunization, 1:1 OVA+ and OVA− target cells were CFSE-labeled and adoptively transferred to measure the endogenous CTL response against OVA-expressing cells. As expected, since IgHEL mice

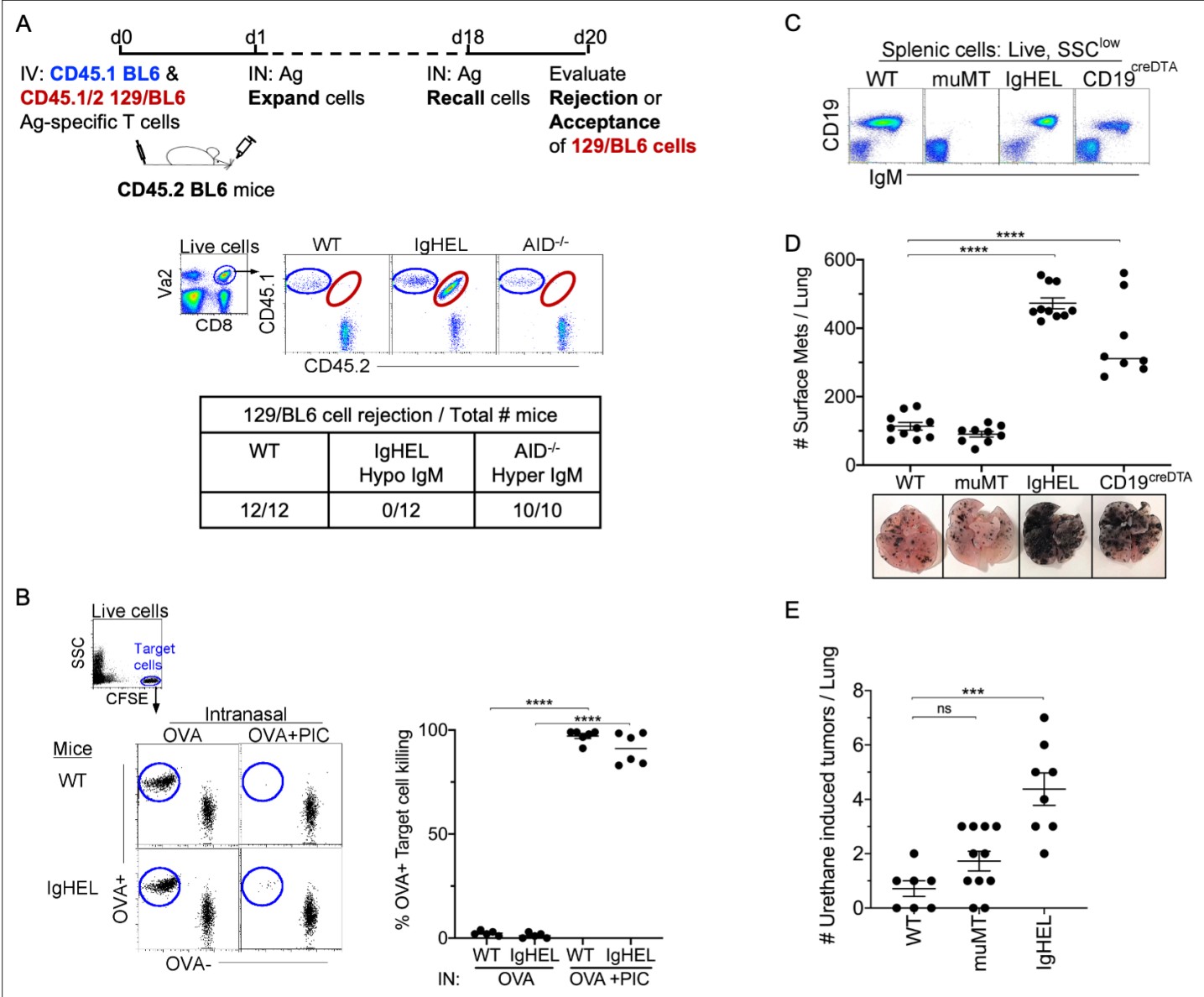

**Figure 1.** NAb repertoire is required for elimination of neoantigen expressing cells in absence of PAMPs. (**A**) *Experimental design*—Splenocytes from CD45.1 BL6-OT-I (internal control) and CD45.1/2 129BL6-OT-I were adoptively transferred into CD45.2 WT, IgHEL, and *Aicda* mice followed by i.n. delivery of OVA (Ag), which expands the cells expressing 129 neoantigens for detection in BL6 host. The mice were rechallenged with OVA on day 18 to assess 129BL6 cell rejection at day 20. Table illustrates # rejected/mice examined. (**B**) WT and IgHEL mice were instilled with 2 µg of OVA alone or with 10 µg of poly I:C. At day 6, CFSE-labelled OVA⁻ and OVA⁺ target cells were transferred and assessed killing at day 8. Flow illustrates CFSE⁺ target cells plotted as OVA⁺ and OVA⁻ cells. Dots represent the number of mice. (**C**) Flow analysis of circulating lymphocytes in WT, muMT, IgHEL, and CD19creDTA mice plotted as CD19 versus IgM (**D**) WT, muMT, IgHEL, and CD19creDTA mouse lungs were inflated 16 days after i.v. B16F10 challenge. Pics depict total surface metastases (mets) per lung, which were enumerated and illustrated by scatter plot, each dot represents one mouse. Combined data of two independent experiments with 4–5 mice per group. ****$p<0.0001$. (**E**) WT, IgHEL, and muMT mouse lungs were inflated 6 months after urethane injections (*Figure 1—figure supplement 1B*). Scatter plot, each dot represents one mouse. ***$p<0.0001$, ns-non significant, mean ± SEM. NAb, natural antibody; PAMP, pathogen-associated molecular pattern; WT, wild-type.

The online version of this article includes the following figure supplement(s) for figure 1:

**Figure supplement 1.** Tumor burden is greater in antibody-deficient mice.

exhibit normal DC migration from the tissue to the draining lymph nodes (*Atif et al., 2018*), a similar CTL response against OVA-expressing cells was observed when WT and IgHEL mice were immunized with OVA and poly I:C (*Figure 1B*).

Next, we examined whether NAbs are required for anti-tumor immunity using the B16F10 melanoma model. Three distinct B cell-deficient mouse models were challenged with melanoma. The first was the most popular B cell-deficient mouse model, muMT mice, which lacks CD19+ B cells. muMT mice did not show enhanced tumorigenesis above WT counterparts, as previously reported (*Figure 1C and D*). The second strain tested was IgHEL mice, where greater than >90% of B cells are IgM-specific for hen egg lysozyme (*Figure 1C*). The third strain was CD19^creDTA mice, which have 70–90% less B cells, predominantly due to the loss of B1 cells, compared to WT mice. Both strains are thus B cell-deficient, and both showed an approximately three- to fivefold increase in tumor burden compared to WT mice (*Figure 1D*, *Figure 1—figure supplement 1A*; *Atif et al., 2018*). Similar observations were made of a spontaneous, chemically induced lung cancer model, urethane (*Figures 1E* and *Figure 1—figure supplement 1B*). The observation that two distinct B cell-deficient mice, compared to the traditionally used muMT mouse, showed enhanced tumor burden suggests that antibody-producing B cells significantly contribute to anti-tumor immunity.

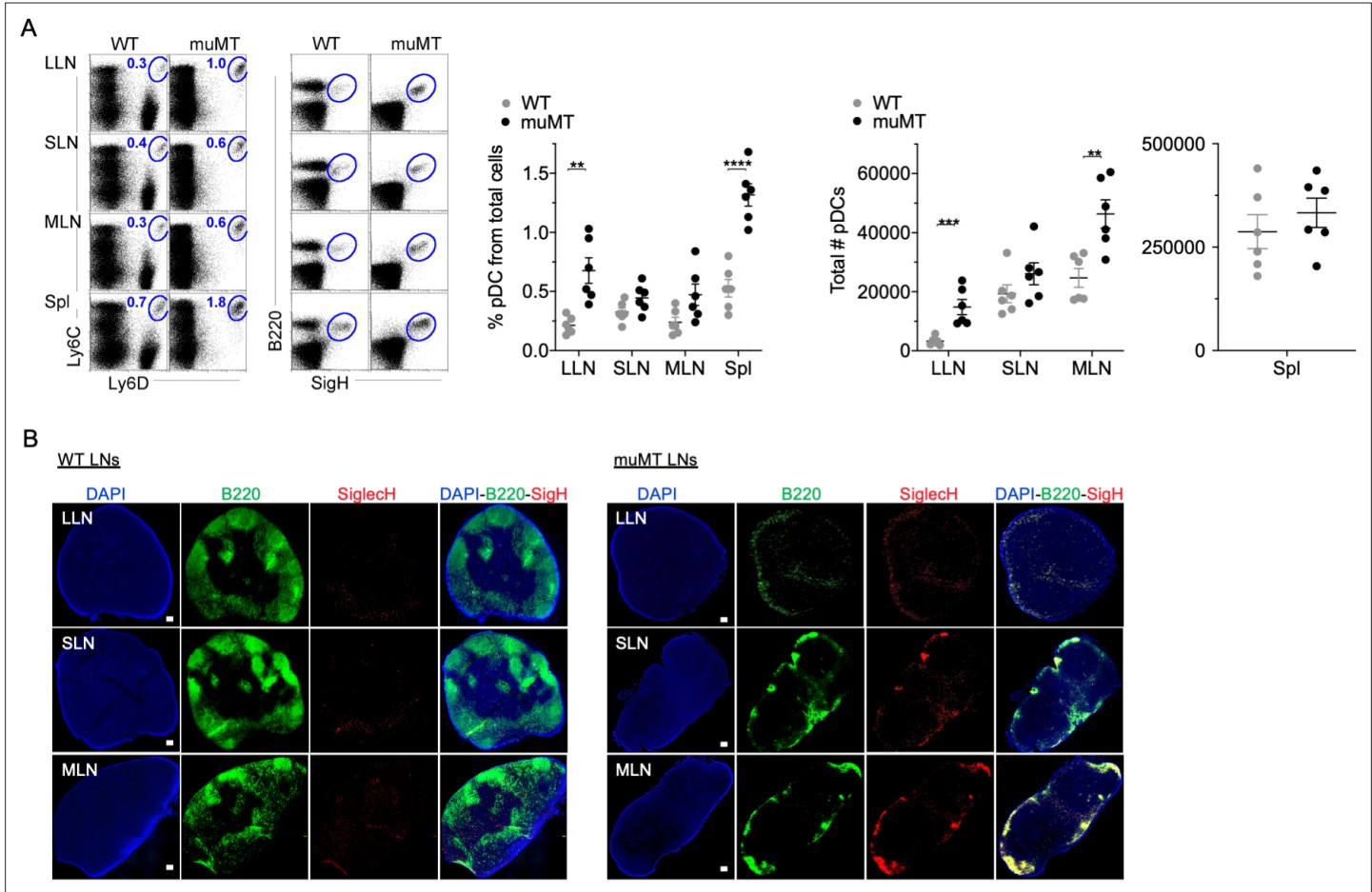

**Figure 2.** muMT mice have elevated levels of pDCs in lymph nodes and spleen. (**A**) Left, flow cells plotted as Ly6C versus Ly6D and B220 versus SiglecH to identify pDCs from B cells in lung draining-LNs (LLN), skin draining-LNs (SLN), mesenteric draining-LNs (MLN), and Spleen (Spl). Right, scatter plot displays the pDC frequency and count in WT and muMT mice. Data are representative two combined out of four independent experiments with 3 mice per group. \*\*p<0.0097, \*\*\*\*p<0.0001, Gating strategy, *Figure 2—figure supplement 1*. (**B**) IHC of LLN, SLN, and MLN from naïve WT and muMT mice. Sections were stained with DAPI (blue), anti-B220 (green), and anti-SiglecH (red). Scale bars, 100 μm. Data are representative of three independent experiments. IHC, immunohistochemistry; pDC, plasmacytoid dendritic cell; WT, wild-type.

The online version of this article includes the following figure supplement(s) for figure 2:

**Figure supplement 1.** Flow plots illustrate pDC stains and gating strategies.

## muMT mice have excess plasmacytoid dendritic cells (pDCs), with most located in the absentee B cell zone

The disparity in tumor burden between the muMT and IgHEL mice led us to probe whether muMT mice show abnormalities in other immune cell types that could compensate for B cell deficiency, particularly mononuclear phagocytes and pDCs. We did not observe any substantial difference in mononuclear phagocyte populations between muMT and WT mice (data not shown). However, the numbers of pDCs, which are lymphoid in origin (*Sathe et al., 2013*; *Chang et al., 2015*; *Rodrigues et al., 2018*; *Dress et al., 2019*), were significantly greater in the muMT mice compared to WT mice (*Figure 2A*). PDCs were identified in LNs and spleen by their co-expression of either Ly6C versus Ly6D, or B220 versus SiglecH (*Figures 2A* and *Figure 2—figure supplement 1*). B cells only express Ly6D, but not Ly6C or SiglecH (*Figure 2A*). Interestingly, all B220+ cells in the muMT mice were SiglecH+, supporting the notion that these cells are not residual B cells in the muMT mice, but instead pDCs (*Figure 2A*).

Next, we examined where the increased number of pDC were located in LNs by immunohisto-chemistry (IHC). B220+SiglecH+ pDCs were observed in the paracortex and medullary region of WT and muMT mice (*Figure 2*, *Figure 3—figure supplement 1*). However, a marked difference was the unique clustering of pDCs in muMT mice within the 'absentee B cell zone,' a region adjacent to the subcapsular sinus (*Figure 2B*). This clustering occurred even in the absence of CXCR5 expression on pDCs (*Figure 3—figure supplement 1B*). Since pDCs are potent anti-viral and anti-tumor immune cells (*Mitchell et al., 2018*; *Poropatich et al., 2020*), we hypothesized that in the muMT mice pDCs are compensating for B cells to elicit an anti-tumor immune response.

## Diminishing pDCs numbers in muMT highlighted the significance of NAbs in promoting anti-tumor immunity

PDCs could either promote or suppress tumor progression depending on their activation state (*Mitchell et al., 2018*). IFN-α, expressed by pDCs, is a robust anti-tumor cytokine. PDCs are also known as type 1 IFN-producing cells, as they produce type 1 IFNs in larger quantities compared to other immune cell types (*Diebold et al., 2004*; *Dunn et al., 2005*; *Ito et al., 2005*). Therefore, we examined by IHC whether pDCs were producing IFN-α in the tumor microenvironment. We made two observations. First, there were significantly more pDCs in the tumor microenvironment of muMT mice than WT mice. Second, these pDCs expressed significant levels of IFN-α compared to other cells in the surrounding tissue (*Figure 3A*, *Figure 3—figure supplement 1*). Thus, an increased influx of IFN-α producing pDCs in the tumor might explain why muMT mice do not show enhanced tumor burden in the absence of B cells.

To determine whether the overabundance of pDCs was promoting anti-tumor immunity in muMT mice, we selectively diminished ~70–90% of tissue pDCs using a triple antibody cocktail (for details, Supplementary materials and *Figure 3—figure supplement 2*). Although the literature suggests that any one of the three antibodies could deplete B220+ or SigH+ cells, we found that effective depletion of pDCs required the triple antibody cocktail. The use of only one or two resulted in the masking of the epitope but not pDC depletion, which was revealed using Ly6C versus Ly6D (*Figure 3—figure supplement 2*). Depleting pDCs in muMT mice resulted in a threefold increase in tumor burden and size compared with non-depleted muMT and WT mice (*Figure 3B* and *Figure 3—figure supplement 3A*). Moreover, reconstitution of NAbs with naive WT serum in tumor-bearing pDC-depleted muMT mice significantly diminished the tumor growth compared to pDC-depletion alone (*Figure 3*, *Figure 3—figure supplement 3*). This underscores the role that NAbs play in anti-tumor immunity.

In WT mice, pDCs in the tumor microenvironment recruit and activate NK cells (*Joao et al., 2004*; *Liu et al., 2008*; *Persson and Chambers, 2010*; *Swiecki et al., 2010*). In addition, type I IFN signaling has been associated with NK cell-mediated tumor surveillance (*Dunn et al., 2005*). Unlike IFN-γ, which directly target tumor cells, type I interferons such as IFN-α facilitate anti-tumor immunity through the activation of immune cells such as NK cells and CD8 T cells (*Dunn et al., 2005*). Since pDCs and IFN-α were prevalent in muMT mice, we next assessed the frequency of NK cells. As anticipated, tumor-bearing muMT mice had significantly more NK cells in their tumor microenvironment than WT mice. Moreover, NK cell frequency in the tumors of muMT mice were diminished after pDC depletion (*Figure 3C*). These findings suggest that excess pDCs mediate the recruitment of NK cells in muMT mice. Finally, we investigated whether neutralizing IFN-α, which is highly secreted by pDCs

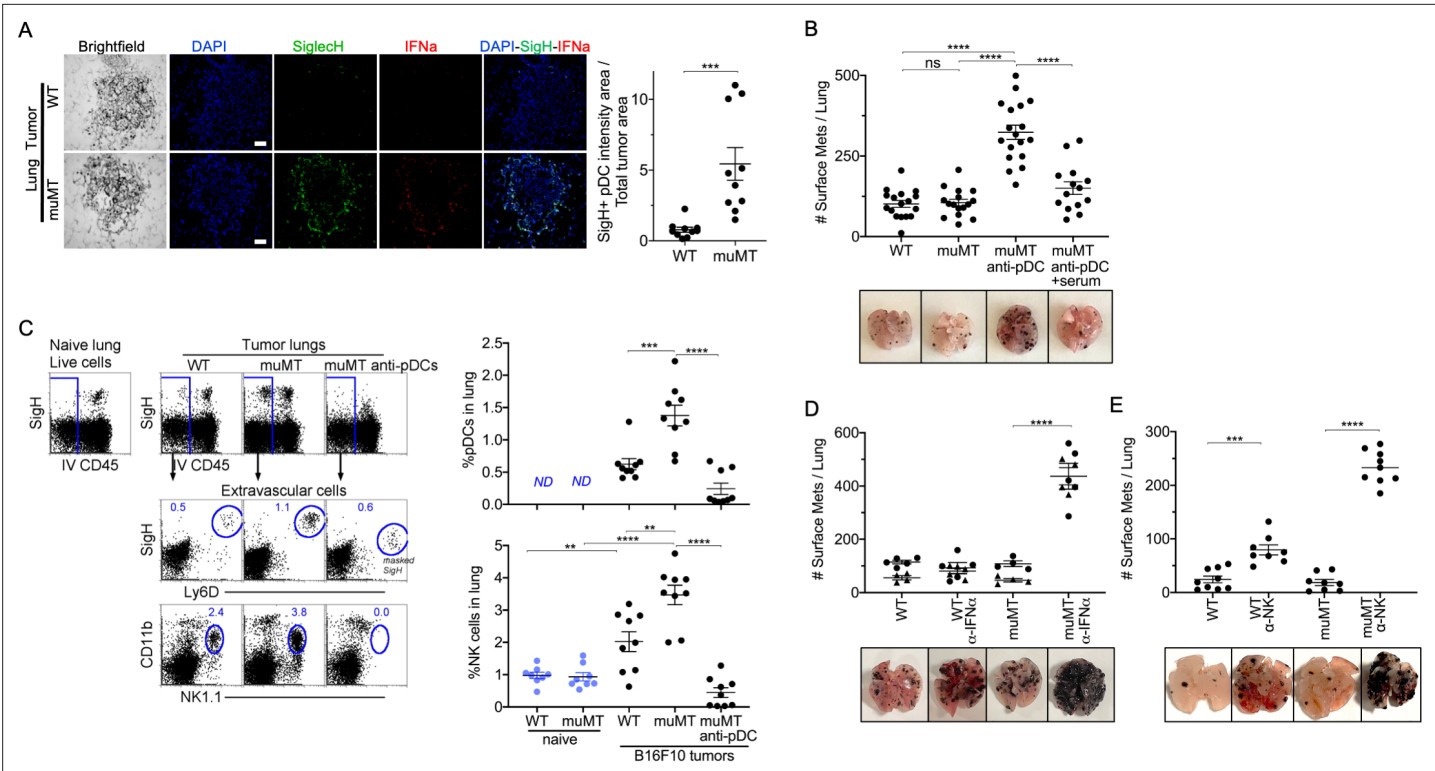

**Figure 3.** Depletion of pDCs in muMT mice results in increased tumor burden and decreased infiltration of NK cells at the tumor site. (**A**) IHC of lung B16F10 melanoma in WT and muMT mice stained with DAPI (blue), anti-SiglecH (green), and anti-IFN-α (red). Scale bars, 40 µm. Scatter plot illustrates Keyence quantitative software analysis of SigH⁺ intensity over total tumor area×100. Each dot plot represents one random tumor analyzed per mouse from 5 mice per group. Data are representative of three independent experiments. (**B**) WT and muMT mouse lungs were inflated 16 days after i.v. B16F10 melanoma cell. Pics depict total surface metastases (mets) per lung, which were enumerated and illustrated by scatter plot, each dot represents one mouse. Two muMT groups were pDC-depleted, one group was given naive WT serum. Combined data of three independent experiments with 4–5 mice per group. ****p<0.0001. Non-significant (ns). Mean ± SEM. Anti-pDC treatment—***Figure 3—figure supplement 2***. Tumor burden— ***Figure 3—figure supplement 3***. (**C**) At day 16, B16F10 tumor-induced WT and muMT mice were iv injected with anti-CD45 before harvesting, to exclude intravascular cells. Gating strategy: Top row, flow plots, gated on live cells followed by extravascular cell analysis (negative cells, iv CD45) in the naive and tumor-bearing lungs. Tumor cells plotted in second and third rows were gated on pDCs, SiglecH⁺Ly6D⁺, and NK cells CD11b⁺NK1.1⁺. Right, scatter plots display the pDCs and NK cell frequency in tumors of WT and muMT **p<0.0040, ***p<0.0077, ****p<0.0001, *ND* (not detected). Anti-pDC treatment—***Figure 3—figure supplement 2***. (**D**) Pics depict total surface metastases (mets) per lung in WT and muMT mice with and without anti-IFN-α treatment, each dot represents one mouse. ****p<0.0001, mean ± SEM. (**E**) Scatter plot represents individual mice from WT and muMT mice treated with and without anti-NK1.1. **p<0.001, ****p<0.0001, mean ± SEM. Data combined from two independent experiments with 3–5 mice per group. IHC, immunohistochemistry; pDC, plasmacytoid dendritic cell; WT, wild-type.

The online version of this article includes the following figure supplement(s) for figure 3:

**Figure supplement 1.** pDC staining of WT and muMT lymph nodes and lung tumors.

**Figure supplement 2.** Depletion of the anti-tumor compensatory cell type, pDCs, in muMT mice.

**Figure supplement 3.** Anti-pDC treatment alone increased the tumor burden in muMT mice, but anti-pDC treatment given with naive WT serum reduces the tumor burden.

in muMT mice, or depleting NK cells would increase the tumor burden. Indeed, low concentration of neutralizing antibodies against IFN-α significantly increased the tumor burden in muMT mice but not in WT mice (***Figure 3D***). Moreover, depletion of NK cells in muMT mice resulted in a fivefold increase in tumor burden compared to depletion of NK cells in WT mice, suggesting that the pDC-IFN-I-NK cell axis is the main anti-tumor immune mechanism in place for muMT mice. In contrast, WT mice display other anti-tumor immune mechanisms since the tumor burden after the depletion of NK cells was not as dramatic as those observed for muMT mice (***Figure 3***, ***Figure 3—figure supplement 3***). Taken together, these findings indicate that the compensatory mechanism exhibited in the muMT mice derives from the development of pDCs, which are recruited to the tumor microenvironment in

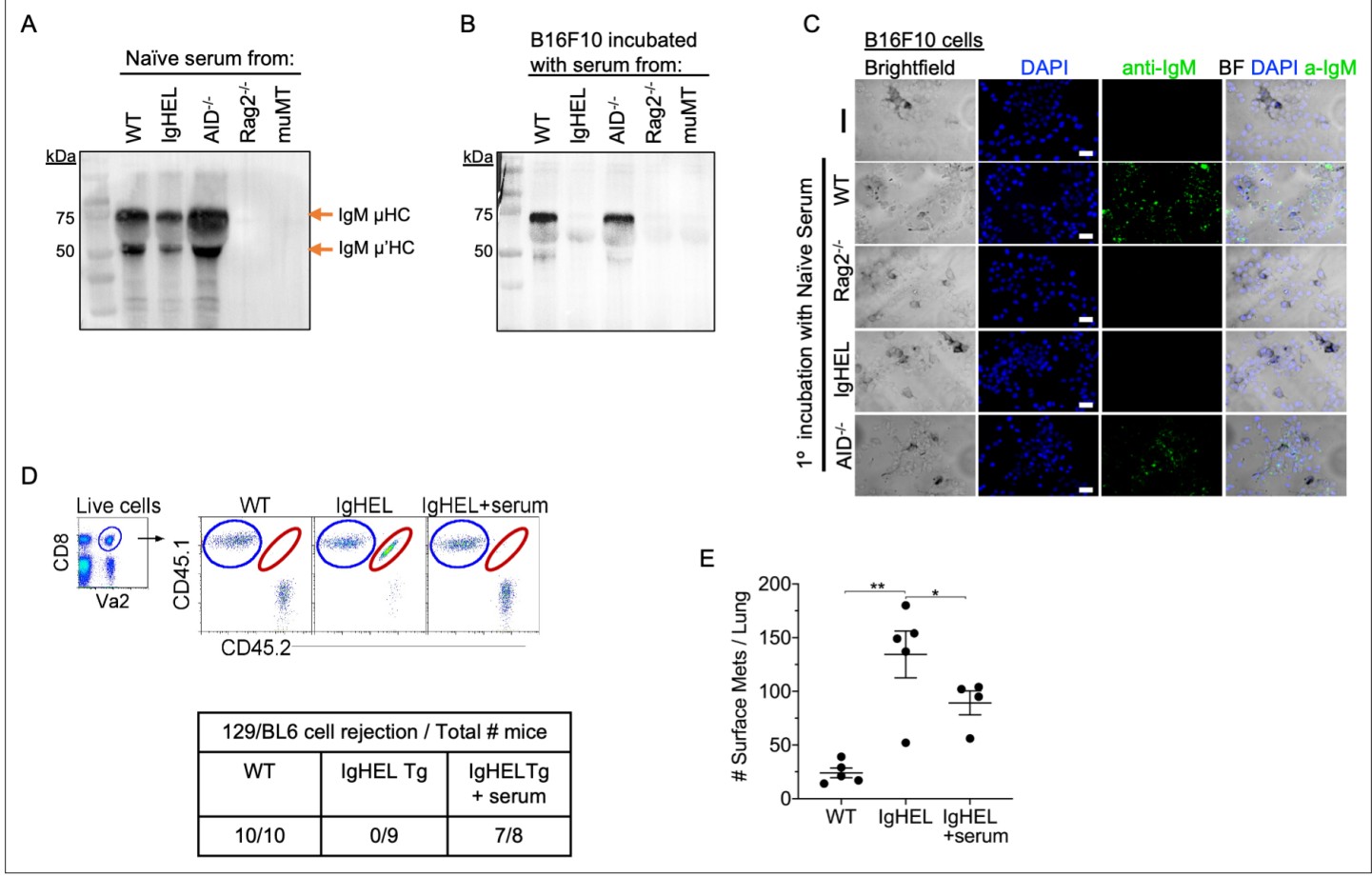

**Figure 4.** Natural IgM repertoire tags neoantigen expressing cells for clearance. (**A**) Western blot IgM analysis of naive serum from WT, IgHEL, *Aicda*, *Rag2⁻/⁻*, and muMT mice *Figure 4—source data 1*. (**B**) B16F10 cells preincubated with naïve WT, IgHEL, *Aicda*, *Rag2⁻/⁻*, and muMT serum, washed and run by Western blot for IgM analysis *Figure 4—source data 1*. (**C**) Live B16F10 cells were plated on Lab-Teks slides and incubated with either naïve WT, *Rag2⁻/⁻*, IgHEL, *Aicda*, and muMT serum or no serum, to detect IgM binding on cells (green). Scale bars, 40 µm. (**D**) *Experimental design* as in *Figure 1A*. C57BL/6 WT, IgHEL, and IgHEL mice treated with serum were analyzed for the rejection of CD45.1/2 129BL6 cells. Table illustrates # rejected/mice examined. (**E**) Scatter plot illustrates the number of B16F10 induced surface mets in WT and IgHEL mice, each dot represents one mouse. *p<0.01, **p<0.003. WT, wild-type.

The online version of this article includes the following source data for figure 4:

**Source data 1.** Western blot analysis of B16F10 serum binding.

significant numbers, secrete high levels of type 1 IFN, and recruit and activate NK cells that promote anti-tumor immunity.

## Natural IgM forms an immune complex with neoantigen expressing cells, leading to their elimination

Our data suggest the possibility of an immune complex formation between natural IgM antibodies and nascent tumor cells. To demonstrate the crosslinking of IgM with B16F10 melanoma, we seeded B16F10 cells on Lab-Teks and added fresh serum from naïve WT, IgHEL,c*Aicda*, *Rag2⁻/⁻*, and muMT mice for 30 min. Afterward, cells were either washed and extracted for Western blot analysis or counter stained with a secondary antibody to detect IgM binding to live cells (*Figure 4A-C*). Western blot analysis and IHC imaging showed that NAbs from WT and *Aicda* mice bind to B16F10 cells, whereas no binding was detected in naive serum derived from IgHEL, *Rag2⁻/⁻*, and muMT mice *Figure 4A-C*. These findings highlight that NAbs recognize and tag neoantigen-expressing cells.

Finally, we determined whether the reconstitution of NAbs in IgHEL mice mediates the rejection of neoantigen-expressing cells using two models: 129/BL6 neoantigen-expressing cells and B16F10 melanoma models. As previously observed, IgHEL mice do not reject 129/BL6 cells (*Figure 4D*). In

contrast, WT and IgHEL mice reconstituted with naive WT serum completely rejected the adoptively transferred 129/BL6 cells (*Figure 4D*). In the melanoma model, the tumor burden of IgHEL mice given naive WT serum was significantly reduced compared to untreated IgHEL mice but was still higher than in WT mice (*Figure 4D*). Overall, these data support the hypothesis that B cells secreting NAbs are critical for detection and elimination of nascent cancer cells—unless abnormal compensatory mechanisms develop, such as increased tumor infiltration by IFN-I producing pDCs and NK cells in muMT mice.

## Discussion

When the immune system encounters pathogens, innate immune cells—including NAbs—are thought to be the first line of defense. It is thus reasonable to hypothesize that a similar line of defense may exist for precancerous cells that recently escaped intrinsic tumor suppression mechanisms. The early recognition of precancerous cells by NAbs would allow the immune system to use multiple components of the innate branch prior to T cell activation, which is part of the adaptive branch. Our findings support the innate immune hypothesis. We found marked increases in tumor burden in three distinct B cell-deficient mouse models—including muMT mice after pDC depletion—and after B cell-derived NAb depletion. Reconstituting NAbs in deficient mice was sufficient to restore anti-tumor immunity. These findings are consistent with an innate immune response in which we envision (requiring further investigation) that NAbs tag nascent, precancerous cells for clearance, followed by activation of complement, engulfment by C1q secreting interstitial macrophages (which are present in all tissues) (*Gibbings et al., 2017*; *Leach et al., 2020*), surveying monocytes (*Jakubzick et al., 2013*), and NK cell antibody-dependent cellular cytotoxicity.

Although muMT mice possess many immune cell abnormalities, such as reduced T cell numbers, an absence of follicular DCs, poor lymphatic development and DC migration, and small spleens (*Asano and Ahmed, 1996*; *Crowley et al., 1999*; *Angeli et al., 2006*; *Schreiber et al., 2011*), these mice were used to define for the role of B cells in anti-tumor immunity for decades. Although this study does not challenge the empirical validity of previous findings in muMT mice, which are reproducible, it highlights that relying on results from a single mouse model delayed the discovery of the fundamental role B cells play in both innate and adaptive anti-tumor immunity.

Our findings raise the question of why B cell-deficient muMT mice exhibit increased pDC frequencies, which compensate for B cell deficiency and permit anti-tumor immunity. pDCs and B cells share a common lymphoid progenitor (*Rodrigues et al., 2018*; *Dress et al., 2019*). The genetic abnormality that makes the progenitor unable to differentiate into B cells in this model may permit differentiation into pDCs instead. As pDCs normally represent a small fraction of immune cells in the lymphoid compartment in WT mice, they may not be present in sufficient numbers to enter into the tumor environment in WT mice. In muMT mice, pDCs are present in much higher numbers, and this abundance appears to enable sufficient numbers to migrate into the tumor microenvironment to mount an anti-tumor response. We also observed clustering of pDCs in the absentee B cell zone in muMT mice, but not in WT mice (*Figure 2B*). The implications of pDCs residing in the absentee B cells zone of the LNs are unclear.

These findings also converge with previous results on NAb-deficient IgHEL mice. Previously, we showed that IgHEL mice developed significantly more tumors than mice deficient in DC1, a potent cross-presenting DC (*Atif et al., 2018*). However, DC1-deficient mice develop more tumors than WT mice. This observation suggests that for anti-tumor immune responses, NAbs contribute to both innate and adaptive immunity, whereas DC1 mainly contributes to adaptive immunity, and in particular the activation of cytotoxic T cells. Therefore, it is not surprising that a mouse deficient in both innate and adaptive responses against tumors (IgHEL mice) would develop more tumors than a mouse deficient in only adaptive immunity. However, this hypothesis may also depend on the site of tumor development, such as B16F10 injections in the skin may develop comparatively more DAMPs by passing the need for NAbs to mount anti-tumor immunity.

In summary, our findings indicate that a complete NAb repertoire is required to identify altered 'self' and initiate the elimination of neoantigen-expressing cells, resulting in anti-tumor immunity. This conclusion is supported by converging evidence from multiple B cell and NAb-deficient mice paired with NAb repertoire reconstitution. The implication is that innate immunity is a critical part of cancer immunosurveillance and clearance of precancerous cells—in short, that NAbs are the 'keys' that start

the 'engine' of both innate and adaptive immune responses (*Atif et al., 2018*). More broadly, our findings indicate that cancer elimination begins with an innate immune response, as with other pathogens and injuries.

# Materials and methods

## Mice

CD45.2 WT (000664), C57BL/6 Ly5.1 (002014), OT-I transgenic (003831), 129S1/SvlmJ (002448), Act-mOVA (005145), B6.129S2-*Ighm*$^{tm1Cgn}$/J (muMT, 002288), *Aicda* (AID$^{-/-}$, 008825), IghelMD4 (IgHEL, 002595), *Rag2*$^{-/-}$ (008449), B6.129P2(C)-*Cd19*$^{tm1(cre)Cgn}$/J (CD19cre, 006785), and B6.129P2-*Gt(RO-SA)26Sor*$^{tm1(DTA)Lky}$/J (Rosa-DTA, 009669) mice were purchased from Jackson Research Laboratories and Charles River NCI. *Aicda* mice (hyper-IgM), B cells are unable to undergo Ig class switch recombination and have only serum IgM in circulation, lacking all other Ig isotypes (*Muramatsu et al., 2000*). In IgHEL mice (hypo-IgM) greater than 90% of IgM-secreting B cells are specific for hen egg lysozyme. All mice were bred in house. Mice were genotyped or phenotyped prior to studies and used at 6–8 weeks of age, housed in a specific pathogen-free environment at Dartmouth Hitchcock Medical College, an AAALAC accredited institution, and used in accordance with protocols approved by the Institutional Animal Care and Utilization Committee.

## Flow cytometry

Tissues were minced and digested with 2.5 mg/ml collagenase D (Roche) for 30 min at 37°C. 100 µl of 100 mM EDTA was added to stop 1 ml of enzymatic digestion. Digested tissue was pipetted up and down 30 times using a glass Pasteur pipette and passed through a 70 µm nylon filter to acquire single-cell suspensions from lungs, spleen, lung draining-LNs, skin draining-LNs, and mesenteric draining-LNs. Cells were stained with the following monoclonal Abs: Phycoerythrin (PE)-conjugated to Siglec H and NK1.1; PerCP-Cy5.5-conjugated to B220; PE-Cy7-conjugated to CD11c and CD44; BUV395-conjugated to CD11b and CD4; fluorescein isothiocyanate (FITC)-conjugated to Ly6D, CD3, CD27, and CD62L; allophycocyanin-conjugated to CD19 and CD103; APC-Cy7-conjugated to CD45; and BV510-conjugated to Ly6C. The viability dye DAPI (#D9542, Sigma-Aldrich) was added immediately before each sample acquisition on a BD Symphony A3 analyzer (BD Biosciences). Data were analyzed using FlowJo (Tree Star, Ashland, OR). Antigen-specific antibodies and isotype controls were obtained from BioLegend, eBioscience, and BD Biosciences.

## 129 neoantigen rejection model

129 neoantigen rejection model: C57BL/6 CD45.1 OT-I mice were crossed with 129Sv mice to create an F1 129/BL6 OT-I mouse. 129 strain has allelic variations outside of the MHC locus of C57BL/6 strain. Female 129/BL6 OT-I cells were used to introduce neoantigens into female C57BL/6 mice. Model set up: two million CD45.1/2 129/BL6 OT-I cells and CD45.1 BL6 OT1 cells were transferred intravenously into congenic recipients. About 2 mg freshly prepared and filtered ovalbumin (OVA) was given, resulting in the expansion of adoptively transferred neoantigen-expressing T cells. Mice were then rechallenged with 100 mg OVA at day 18 to recall adoptively transferred cells. At 2 days after re-challenge, the lung-draining LNs were examined for the presence (recall) or absence (rejection) of adoptively transferred neoantigen-expressing cells. During this experiment, the cages were changed every 72 hr to minimize the mice exposure to any possible PAMPs.

## Cell lines

B16F10 melanoma cells (CRL-6475) and LLC1 tumor cells (CRL-1642; ATCC) were purchased from ATCC and maintained in RPMI with 10% FCS, 1% Pen/Strep/L-glutamine (Sigma-Aldrich), 1% non-essential amino acids (Sigma-Aldrich), 1% sodium pyruvate (Sigma-Aldrich), 10 mM HEPES (Sigma), and 0.1 mM β-mercaptoethanol. The cell lines were confirmed free of mycoplasma contamination and the identification was authenticated through STR DNA profiling.

## B16F10 lung melanoma model

Mice were intravenously challenged with $2\times10^{5}$ viable B16F10 cells and euthanized 16 days post-injection. Lungs of mice were inflated with 1% agarose. A blinded observer counted the B16F10

lung surface metastases. Tumor analysis represented in this study use female mice. However, similar observations were made in male mice.

### Lewis lung carcinoma induced tumors

Mice were injected with $6.0 \times 10^5$ LLC1 tumor cells intravenously and euthanized on day 21. The lungs were perfused with cold 1% phosphate-buffered saline (PBS) and then inflated with Agarose mixed with India Ink and dipped in Fekete's solution. Tumor counts were performed on both dorsal and ventral sides of the lungs.

### Plasmacytoid dendritic cells depletion in muMT mice and IFN-α neutralization

To deplete pDCs in muMT mice, 1 mg of anti-Siglec H (clone 440c Rat IgG2b, BioXcell), 1 mg anti-mouse B220, (clone RA3.3A1/6.1 Rat IgM, BioXcell), and 1 mg anti-B220 (clone RA3-6B2 Rat IgG2a, Leinco technologies, Inc) were injected individually and in combination into the peritoneal cavity of the mice. Assessment of pDC depletion from the spleen, LLN, SLN, and MLN was performed at 24, 48, and 72 hr. To deplete pDCs long-term in the B16F10 melanoma model, mice were i.p. injected with the triple Ab treatment every 48 hr until harvest, day 16. A group of pDC-depleted mice was also i.p injected with 200 µl of native WT serum every 48 hr until lung harvest.

To neutralize IFN-α in WT and muMT mice, 100 µg of anti-IFN-α (clone TIF-3C5 Armenian Hamster IgG, Leinco Technologies, Inc) was injected intraperitoneally in B16F10 injected mice every 48 hr until harvest, day 16.

### NK cell depletion

To deplete NK cells in WT and muMT mice, 50 µg of anti-NK1.1 (clone PK136 Mouse IgG2a, BioXcell) was injected intraperitoneally in B16F10 injected mice every 72 hr until harvest, day 16.

### Urethane model

Spontaneous lung tumors were induced with intraperitoneal injections of 1 mg/g urethane (ethyl carbamate; Sigma-Aldrich) weekly for 6 weeks. Mice were euthanized 6 months after the last intraperitoneal injection of urethane. Hematoxylin and eosin staining was performed on the lungs and tumors were enumerated and statistically analyzed.

### Microscopy

Lungs and LNs were excised and immersed in 4% paraformaldehyde with 10% sucrose and 7% picric acid in PBS for 2 hr and then embedded in Tissue-Tec OCT (Thermo Fisher Scientific). 10 µm sections were prepared and slides were stained for B220, SiglecH, and IFN-α. Control slides were treated with Ab isotype controls and secondary antibodies. Stained slides were mounted with Prolong Gold Antifade containing DAPI (# P36931, Invitrogen), whole lymph node images were stitched using 10× objective lens and imaged with a Keyence fluorescence microscope (BZ-X800 series).

### Statistics

Statistical analysis was conducted using InStat and Prism software (GraphPad). All results are expressed as the mean ± SEM. Statistical tests were performed using a two-tailed Student's t-test. A value of $p < 0.05$ was considered statistically significant.

## Acknowledgements

Thank you to Dr. Raul Torres for insightful discussions in B cell biology and Dr. Bart Lambrecht for suggesting the CD19creXRosaDTA mouse model as an alternative model for B cell deficiency. This work was supported by the National Institutes of Health grants R01 HL115334, R01 HL135001, and R35 HL155458 (CVJ). CVJ, TW, and KR prepared the manuscript; KR, AT, MJM, TW, and CVJ executed the experiments; all authors provided intellectual input, critical feedback, discussed results, and designed experiments.

## Additional information

### Funding

| Funder | Grant reference number | Author |
|---|---|---|
| National Heart, Lung, and Blood Institute | R01 HL115334 | Claudia Jakubzick |
| National Heart, Lung, and Blood Institute | R01 HL135001 | Claudia Jakubzick |
| National Heart, Lung, and Blood Institute | R35 HL155458 | Claudia Jakubzick |

The funders had no role in study design, data collection and interpretation, or the decision to submit the work for publication.

### Author contributions

Kavita Rawat, Conceptualization, Data curation, Investigation, Methodology, Writing – original draft, Writing – review and editing; Anita Tewari, Investigation, Methodology; Madeline J Morrisson, Writing – review and editing; Tor D Wager, Data curation, Methodology, Writing – review and editing; Claudia V Jakubzick, Conceptualization, Data curation, Funding acquisition, Investigation, Methodology, Resources, Supervision, Validation, Visualization, Writing – review and editing

### Author ORCIDs

Claudia V Jakubzick http://orcid.org/0000-0002-3731-0198

### Ethics

The mice were housed in a specific pathogen-free environment at Dartmouth Hitchcock Medical College, an AAALAC accredited institution, and used in accordance with protocols approved by the Institutional Animal Care and Utilization Committee of Dartmouth College (#00002229a). The institutional welfare assurance number is A3259-01.

### Decision letter and Author response

Decision letter https://doi.org/10.7554/eLife.69713.sa1
Author response https://doi.org/10.7554/eLife.69713.sa2

## Additional files

### Supplementary files

• Transparent reporting form

• Supplementary file 1. Articles not supporting and supporting the hypothesis that B596 cells are essential for early cancer cell recognition and anti-tumor immunity. *As of597 February 2021.

### Data availability

All data generated or analysed during this study are included in the manuscript and supporting files. Source data files has been provided for Figure 4.

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
