## [Decision Letter]

**Acceptance summary:**

This manuscript presents data that collectively suggest that the innate recognition of neoantigen-expressing cells by natural antibodies may be essential for the early elimination of cancer cells. Overall, this paper is important as it brings to light a hitherto unappreciated mechanism of cancer elimination, that should be considered with further studies.

**Decision letter after peer review:**

Thank you for submitting your article "Re(de)fining innate natural antibodies as contributors to the first line of defense against precancerous cells" for consideration by *eLife*. Your article has been reviewed by 2 peer reviewers, including Lynne-Marie Postovit as the Reviewing Editor and Reviewer #1, and the evaluation has been overseen by Betty Diamond as the Senior Editor. The following individual involved in review of your submission has agreed to reveal their identity: Caetano Reis e Sousa (Reviewer #2).

Essential revisions:

1) The B16F10 model does not represent early tumorigenesis, wherein this mechanism of elimination may be most important. Moreover, it is unclear whether the phenomena observed is related to cancer immunity (as suggested) or transplant immunity (as the B10F10 are likely to differ from the authors' host mice at multiple minor histocompatibility antigens). Given the reliance on systemic injection of mismatched cells and the lack of clarity as to what the antibodies are binding to, these issues remain open. The authors also did not exclude the possibility that DAMPs/PAMPs and even hypoxia may exist in the lesions. Accordingly, the interpretation that the data highlight a role for natural antibodies specifically in cancer immunoediting is premature. Although out of the scope of the current study, it would be important to repeat experiments using a spontaneous model of cancer. At the very least, the manuscript should be edited so that broader conclusions about anti-cancer immunity are removed. The authors should interpret their data conservatively remembering that they are using a transplantable tumor model in which cell rejection can also reflect allo-immunity.

2) The study relies on only one tumour model: B16F10 injected through the tail vein. It would be important to ensure that an additional syngeneic model, ideally with varying levels of neoantigens, is considered.

*Reviewer #1:*

This manuscript presents data that collectively suggest that the innate recognition of neoantigen-expressing cells by natural antibodies may be essential for the early elimination of cancer cells. This is provocative as the dogma, based largely on data from muMT B-cell deficient mice, largely excludes a role for natural antibodies in cancer elimination. Using an experimental metastasis B16F10 model, and two B-cell deficient mouse models, the authors convincingly demonstrate that B-Cell depletion enhances tumour nodule outgrowth and that this can be mitigated, at least in part, with the supplementation of antibodies, with serum derived from wild type mice. They also reveal a potential mechanism to explain discordant results with muMT mice, wherein higher numbers of pDCs compensate for the lack of B cells, enabling a type 1 interferon response in tumours concomitant with NK cell recruitment. Overall, this paper is important as it brings to light a hitherto unappreciated mechanism of cancer elimination, that should be considered with further studies.

1) The study relies on only one tumour model: B16F10 injected through the tail vein. It would be important to ensure that additional models, with varying levels of neoantigens (due to differing mutational burden) are considered. Moreover, this model likely worked as each nodule starts with small tumour burden. Does it also work for a classical primary tumour growth model, and in this case is spontaneous metastasis reduced? In short, additional experiments should be included that consider mutational and overall tumour burden.

2) The B16F10 model does not represent early tumorigenesis, wherein this mechanism of elimination may be most important. Although out of the scope of the current study, it would be important to repeat studies using spontaneous models of cancer. This should be discussed.

3) Please include the sex of the mice used. This would be important to understand the results.

*Reviewer #2:*

The manuscript by Rawat et al. demonstrates the presence of a humoral immune component that decreases implantation and/or growth of allogeneic cells in C57BL/6 mice. Such cells include B16F10 melanoma cells and cells from 129S1 mice that bear minor histoincompatibility differences.

The authors identify moieties in B16F10 cells that can be recognised by serum antibodies. Utilizing several B cell-deficient mouse strains and serum transfer rescue experiments, the authors make a persuasive case for the notion that these natural antibodies can act as barriers to cell implantation/growth in this model. Additionally, the authors delineate an interesting pathway in muMT mice in which the expansion of plasmacytoid cells producing type I IFNs counteracts the loss of natural antibody, thereby preserving relative immunity to B16F10. The authors suggest this compensatory mechanism may have masked the contribution of antibodies/B cells to previous studies of anti-cancer immunity that have relied on muMT mice as a model of B cell-deficiency.

The data presented are clear and of interest. However, the interpretation that the data highlight a role for natural antibodies specifically in cancer immunoediting is premature. Is this cancer immunity or transplant immunity (as the B10F10 are likely to differ from the authors' host mice at multiple minor histocompatibility antigens)? Given the reliance on systemic injection of mismatched cells and the lack of clarity as to what the antibodies are binding to, these issues remain open. Further, the authors frequent allusions to the absence of DAMPs/PAMPs and hypoxia in their experiments is confusing. Is it not possible that intravenous injection alone leads to significant cell damage, exposing or releasing DAMPs? In sum, the authors provide convincing experimental evidence for a humoral barrier to allogeneic transplant but what this means for immune surveillance of cancer in an open question.

The authors are advised to steer away from broader conclusions about anti-cancer immunity and interpret their data conservatively remembering that they are using a transplantable tumor model in which cell rejection can also reflect allo-immunity.

[Editors' note: further revisions were suggested prior to acceptance, as described below.]

Thank you for resubmitting your work entitled Redefining innate natural antibodies as important contributors to anti-tumor immunity" for further consideration by *eLife*. Your revised article has been evaluated by Betty Diamond as the Senior Editor, and a Reviewing Editor.

The manuscript has been improved but there are some remaining issues that need to be addressed, as outlined below:

Please edit the introduction. Specifically, it may be misleading to discuss an absence of DAMPs, which may not represent the situation with transplanted cells or urethane. In addition please refer to DAMPs as "damage-associated molecular patterns" as opposed to danger associated molecular patterns.

---

## [Author Response]

Essential revisions:1) The B16F10 model does not represent early tumorigenesis, wherein this mechanism of elimination may be most important. Moreover, it is unclear whether the phenomena observed is related to cancer immunity (as suggested) or transplant immunity (as the B10F10 are likely to differ from the authors' host mice at multiple minor histocompatibility antigens). Given the reliance on systemic injection of mismatched cells and the lack of clarity as to what the antibodies are binding to, these issues remain open. The authors also did not exclude the possibility that DAMPs/PAMPs and even hypoxia may exist in the lesions. Accordingly, the interpretation that the data highlight a role for natural antibodies specifically in cancer immunoediting is premature. Although out of the scope of the current study, it would be important to repeat experiments using a spontaneous model of cancer. At the very least, the manuscript should be edited so that broader conclusions about anti-cancer immunity are removed. The authors should interpret their data conservatively remembering that they are using a transplantable tumor model in which cell rejection can also reflect allo-immunity.

We added to the manuscript a spontaneous, chemically induced lung cancer model, Urethane (Figure 1E, Figure 1-Figure supplement 1B). For reviewers only, to show support of our hypothesis, we provided another spontaneous model that is currently being complied for different study MMTV-PyMT and IgHEL MMTV-PyMT.

As requested, we edited the manuscript to omit the statements that refer to the absence of DAMPs in our model systems and dampened the concept of how NAbs play a role in early stages of tumorigenesis.

2) The study relies on only one tumour model: B16F10 injected through the tail vein. It would be important to ensure that an additional syngeneic model, ideally with varying levels of neoantigens, is considered.

We added to Figure 1-Figure supplement 1 an additional syngeneic model, LLC. In Figure 1E (Figure 1-Figure supplement 1B), we added a spontaneous model, Urethane.

Reviewer #1:This manuscript presents data that collectively suggest that the innate recognition of neoantigen-expressing cells by natural antibodies may be essential for the early elimination of cancer cells. This is provocative as the dogma, based largely on data from muMT B-cell deficient mice, largely excludes a role for natural antibodies in cancer elimination. Using an experimental metastasis B16F10 model, and two B-cell deficient mouse models, the authors convincingly demonstrate that B-Cell depletion enhances tumour nodule outgrowth and that this can be mitigated, at least in part, with the supplementation of antibodies, with serum derived from wild type mice. They also reveal a potential mechanism to explain discordant results with muMT mice, wherein higher numbers of pDCs compensate for the lack of B cells, enabling a type 1 interferon response in tumours concomitant with NK cell recruitment. Overall, this paper is important as it brings to light a hitherto unappreciated mechanism of cancer elimination, that should be considered with further studies.1) The study relies on only one tumour model: B16F10 injected through the tail vein. It would be important to ensure that additional models, with varying levels of neoantigens (due to differing mutational burden) are considered. Moreover, this model likely worked as each nodule starts with small tumour burden. Does it also work for a classical primary tumour growth model, and in this case is spontaneous metastasis reduced? In short, additional experiments should be included that consider mutational and overall tumour burden.

To validate our findings in the B16F10 lung melanoma model, we used syngeneic model and cell line, Lewis lung carcinoma, LLC. We observed significantly more LLC tumors in IgHEL and CD19^creDTA^ mice compared to WT – added to Figure 1-Figure supplement 1.

2) The B16F10 model does not represent early tumorigenesis, wherein this mechanism of elimination may be most important. Although out of the scope of the current study, it would be important to repeat studies using spontaneous models of cancer. This should be discussed.

We agree that the B16F10 model does not represent early tumorigenesis. Therefore, we thought that the introduction of 129 MHC-matched allogeneic cells is a great model for "perceived neoantigens" by the recipient mouse (similar to male cells in syngeneic female mice), understanding that this model is classically viewed and used for minor antigen transplant rejection. The reason we view this model to study early neoantigen-expression is because 129 female cells or male cells in female mice do not create tumor clusters and hypoxic condition releasing danger signals. Perhaps this is not the best way to study this short-lived event, a stage between escape from intrinsic mechanism and early tumor formation. I respectfully welcome advice from the reviewer on how to address this question experimentally. How does one model early tumorigenesis, the precancerous cells that just escaped intrinsic mechanisms, expressing defined neoantigens in a way that can be tracked and reproducible? The classical models of cancer immunoediting require tumor formation in WT mice as the read out, which is the final stage of tumorigenesis.

We are currently testing other MHC matched cells with less allelic variation (B6/10 and C3H) than the 129 mice and continue to find no rejection in IgHEL mice, while there is rejection in WT mice. However, one could still say that this is allogeneic rejection and not a model for early neoantigen recognition.

3) Please include the sex of the mice used. This would be important to understand the results.

This was included in the methods section. We use female mice for our studies; however, we observe similar trends with male mice in all aspects of our study. Perhaps because the effects observed in IgHEL mice compared to WT mice is so strong that the sex does not play a huge role in the overall outcome.

Reviewer #2:[…] The data presented are clear and of interest. However, the interpretation that the data highlight a role for natural antibodies specifically in cancer immunoediting is premature. Is this cancer immunity or transplant immunity (as the B10F10 are likely to differ from the authors' host mice at multiple minor histocompatibility antigens)? Given the reliance on systemic injection of mismatched cells and the lack of clarity as to what the antibodies are binding to, these issues remain open. Further, the authors frequent allusions to the absence of DAMPs/PAMPs and hypoxia in their experiments is confusing. Is it not possible that intravenous injection alone leads to significant cell damage, exposing or releasing DAMPs? In sum, the authors provide convincing experimental evidence for a humoral barrier to allogeneic transplant but what this means for immune surveillance of cancer in an open question.The authors are advised to steer away from broader conclusions about anti-cancer immunity and interpret their data conservatively remembering that they are using a transplantable tumor model in which cell rejection can also reflect allo-immunity.

Indeed, our study does not unfold the details of the early cancer immunoediting stage where NAbs are targeting precancerous neoantigens for elimination and agree that the B16F10 model does not represent early tumorigenesis. Therefore, we thought that the introduction of 129 MHC-matched allogeneic cells is a great model for "perceived neoantigens" by the recipient mouse (similar to male cells in syngeneic female mice), understanding that this model is classically viewed and used for minor antigen transplant rejection. The reason we view this model to study early neoantigen-expression is because 129 female cells or male cells in female mice do not create tumor clusters and hypoxic condition releasing danger signals. Perhaps this is not the best way to study this short-lived event, a stage between escape from intrinsic mechanism and early tumor formation. I respectfully welcome advice from the reviewer on how to address this question experimentally. How does one model early tumorigenesis, the precancerous cells that just escaped intrinsic mechanisms, expressing defined neoantigens in a way that can be tracked and reproducible? The classical models of cancer immunoediting require tumor formation in WT mice as the read out, which is the final stage of tumorigenesis.

We are currently testing other MHC matched cells with less allelic variation (B6/10 and C3H) than the 129 mice and continue to find no rejection in IgHEL mice, while there is rejection in WT mice. However, one could still say that this is allogeneic rejection and not a model for early neoantigen recognition.

To support our hypothesis, we added a spontaneous, chemically induced lung cancer model, Urethane (Figure 1E, Figure 1-Figure supplement 1B).

[Editors' note: further revisions were suggested prior to acceptance, as described below.]

The manuscript has been improved but there are some remaining issues that need to be addressed, as outlined below:Please edit the introduction. Specifically, it may be misleading to discuss an absence of DAMPs, which may not represent the situation with transplanted cells or urethane. In addition please refer to DAMPs as "damage-associated molecular patterns" as opposed to danger associated molecular patterns.

As requested, we changed the danger associated molecular patterns to damage-associated molecular patterns, and removed any reference to the absence of DAMPs in tumor or neoantigen systems.